# Lemairamin (Wgx-50) Attenuates DSS-Induced Intestinal Inflammation in Zebrafish

**DOI:** 10.3390/ijms25179510

**Published:** 2024-09-01

**Authors:** Ling Zhang, Huiru Liu, Haoyi Zhang, Hao Yuan, Dalong Ren

**Affiliations:** College of Animal Science and Technology, Anhui Agricultural University, Hefei 230036, China; zl@ahau.edu.cn (L.Z.); l1217557467@163.com (H.L.); astomble@stu.ahau.edu.cn (H.Z.); 0809yh@stu.ahau.edu.cn (H.Y.)

**Keywords:** IBD, neutrophils, Wgx-50, DSS, zebrafish

## Abstract

Inflammatory bowel disease (IBD) is a chronic non-specific intestinal inflammatory disease that affects millions of people worldwide, and current treatment methods have certain limitations. This study aimed to explore the therapeutic potential and mechanism of action of lemairamin (Wgx-50) in inflammatory bowel disease (IBD). We used dextran sulfate sodium (DSS)-treated zebrafish as an inflammatory bowel disease model, and observed the effect of Wgx-50 on DSS-induced colitis inflammation. The results of the study showed that Wgx-50 could reduce the expression of pro-inflammatory cytokines induced by DSS and inhibit the recruitment of neutrophils to the site of intestinal injury. Further experiments revealed that Wgx-50 exerted its anti-inflammatory effect by regulating the activation of the Akt pathway. These research findings indicate that Wgx-50 possesses anti-inflammatory activity.

## 1. Introduction

Inflammatory bowel disease (IBD) is a chronic non-specific intestinal inflammatory disease that causes debilitation in the human body, and its etiology is not yet fully understood. The disease mainly includes two subtypes: ulcerative colitis (UC) and Crohn’s disease (CD). UC primarily affects the superficial tissue of the intestines, while CD typically affects the deep tissue of the intestinal wall [1]. Each year, millions of people worldwide are affected by inflammatory bowel disease, and the incidence rate is on the rise [2]. The acute attacks of inflammatory bowel disease can be managed symptomatically with the use of anti-inflammatory drugs to alleviate symptoms. However, the disease currently cannot be completely cured. Current treatment methods for IBD still have certain limitations, including the occurrence of severe adverse reactions from long-term medication use and a high proportion of patients who do not respond to treatment [3,4].

The zebrafish is a model for studying innate immunity in vivo [5]. Its embryos have the optical transparency, enabling the real-time tracking of recruited immune cells using transgenic strains [6]. Moreover, zebrafish and mammals exhibit a high degree of conservation in intestinal development and physiology, making them suitable for studying the fundamental processes of intestinal inflammation and injury [7]. Previously, zebrafish have been extensively used as an evaluation model for chemically induced enterocolitis and inflammatory bowel disease (IBD), and have the potential to serve as a cost-effective colitis animal model for early anti-inflammatory drug development projects [8,9]. The most commonly used IBD chemical models involve the induction of colitis through oral administration of dextran sulfate sodium (DSS) or trinitrobenzene sulfonic acid (TNBS) [10]. Previous studies have shown that zebrafish larvae, when subjected to TNBS and DSS treatment, experience disruption of intestinal homeostasis and subsequent inflammation [9,11,12]. In recent years, DSS has been widely used to induce colitis as ingestion of DSS leads to colonic cell death and the occurrence of inflammation due to the activity of colonic toxins [13]. Moreover, grape exosome-like nanoparticles have been shown to induce intestinal stem cells and protect mice from DSS-induced colitis [14]. The recruitment of immune cells to the site of injury is an important characteristic of the inflammatory response. Neutrophils, as a member of the immune cells, play a crucial role in the body’s defense against external invasions [15]. Neutrophils are the most abundant immune cells in the blood circulation and one of the cell types that arrive earliest at the site of injury or infection [16], and they are also the major markers of intestinal inflammation [17].

Inflammation is an important response in both normal physiology and pathology, playing a critical role in the development of many diseases [18]. Thus, exploring natural compounds with anti-inflammatory activity is of significant research importance. In recent years, lemairamin has garnered attention as a natural compound, and its anti-inflammatory properties have generated widespread research interest [19]. Lemairamin (Wgx-50) is a hydroxylamine compound extracted from Sichuan pepper in China [20], with a chemical structure of N-[2-(3,4-dimethoxyphenyl)ethyl]-3-phenylpropenamide [21]. Most compounds extracted from Sichuan pepper exhibit anti-inflammatory, antibacterial, and antioxidant effects, and play important regulatory roles in neuroinflammation and gastrointestinal inflammation [22]. Previous studies have indicated that Wgx-50 can downregulate the expression of pro-inflammatory cytokines and inhibit NF-κB signaling, thereby alleviating neuroinflammation in Alzheimer’s disease [23]. However, the role of Wgx-50 in DSS-induced colitis and its potential molecular mechanisms are still unclear.

Therefore, in this study, we used sodium dextran sulfate (DSS) immersion to treat 3-day-old zebrafish (3 dpf) to establish a zebrafish colitis model. By measuring the expression of pro-inflammatory cytokines and the recruitment of neutrophils to the site of intestinal inflammation, we investigated whether Wgx-50 has an immunomodulatory effect and further explored its mechanism of action. The results showed that Wgx-50 can reduce the expression of pro-inflammatory cytokines in DSS-induced intestinal inflammation and inhibit the recruitment of neutrophils. Additionally, it was discovered that the molecular mechanism behind the anti-inflammatory effect of Wgx-50 is mediated by the Akt pathway.

In summary, lemairamin, as a natural compound, has potential application value in inflammation regulation. Our research has demonstrated that Wgx-50 can effectively alleviate the development and progression of inflammation by modulating the expression of pro-inflammatory cytokines and activating the Akt pathway. Further studies will contribute to a deeper understanding of the relationship between Wgx-50 and inflammation, providing a theoretical basis and potential drug targets for the development of new anti-inflammatory treatment strategies.

## 2. Results

### 2.1. DSS Promotes Recruitment of Neutrophils and Expression of Pro-Inflammatory Cytokines

To establish a colitis model, we treated 3 dpf transgenic zebrafish Tg(*lyz*:EGFP) with various concentrations of DSS for 72 h to induce colitis (Figure 1A,B). Live imaging of zebrafish larvae was performed using a fluorescence microscope to observe the recruitment of neutrophils to the site of intestinal injury. The results showed that DSS at concentrations of 0.3%, 0.4%, 0.5%, 0.6%, and 0.7% significantly increased the recruitment of neutrophils (Figure 1C,D), with the most significant effect observed with 0.5% DSS. Next, we examined the expression of pro-inflammatory cytokines after treatment with different concentrations of DSS. The results showed a significant increase in the expression levels of pro-inflammatory cytokines (such as *il-1β*, *il-6*, *cxcl8a*, and *tnf-α*) compared to the control group, with the most significant effect observed with 0.5% DSS (Figure 1E–H).

### 2.2. Wgx-50 Can Effectively Decrease the Inflammatory Levels Induced by DSS in Colitis

0.5% DSS can significantly induce colitis. To investigate whether Wgx-50 has a regulatory effect on DSS-induced colitis, we treated the samples with 10 µL/L Wgx-50 for 6, 12, and 18 h, respectively (Figure 2A). The results showed that compared to the DSS group, Wgx-50 suppressed the recruitment of neutrophils to the site of intestinal injury (Figure 2B,C). As the treatment time increased, the number of recruited neutrophils gradually decreased. Furthermore, compared to the DSS group, the expression levels of pro-inflammatory cytokines (such as *il-1β*, *il-6*, *cxcl8a*, and *tnf-α*) were significantly reduced in the Wgx-50 treatment group (Figure 2D–G). The expression levels of cytokines gradually decreased with increasing treatment time, exhibiting a time-dependent reduction. These results indicate that Wgx-50 exerts inhibitory effects on the recruitment of neutrophils and the expression of pro-inflammatory cytokines in the zebrafish colitis model, and this effect is time-dependent, with the most significant effect observed after 12 h of Wgx-50 treatment. Based on this finding, we decided to use a 12 h treatment time for Wgx-50 in the subsequent experiments.

### 2.3. DSS and Wgx-50 Had No Impact on the Growth and Development of Zebrafish

In this experiment, we induced intestinal inflammation in zebrafish using DSS and investigated the impact of Wgx-50 on the inflammation. To determine the lack of pharmacological toxicity in the growth and development of zebrafish larvae at the chosen experimental concentrations, we exposed the zebrafish larvae to DSS for 72 h and treated them with Wgx-50 for 12 h. We evaluated the growth and development of the larvae by monitoring a series of parameters, including tail movement, heartbeat, and body length. The results showed that there were no significant changes in tail movement, heartbeat, and body length in the DSS group and Wgx-50 group compared to the control group (Figure 3A–D). Additionally, we employed a behavior monitor to assess the locomotor activity of zebrafish larvae using the PMR experiment (Figure 3E,F). The results showed that the locomotor activity was significantly reduced in the DSS group and Wgx-50 treatment group compared to the control group (Figure 3G,H). These results suggest that DSS-induced intestinal inflammation has a certain impact on the locomotor activity of zebrafish larvae, but the treatment with Wgx-50 did not effectively restore the activity, and no significant effects were observed on tail movement, heartbeat, and body length. This suggests that Wgx-50 may not have the potential to significantly affect these parameters, and the decreased locomotor activity may be related to other factors induced by intestinal inflammation.

### 2.4. Wgx-50 Has a Protective Effect on the Morphology of Zebrafish Intestines after DSS Treatment

We evaluated the protective effect of Wgx-50 on intestinal morphology after DSS treatment by conducting H&E staining and histopathological examination of zebrafish intestinal sections. We used a DSS-induced zebrafish intestinal inflammation model and employed Wgx-50 as an intervention measure to observe and analyze pathological changes in the intestinal tissue, such as inflammation severity and tissue damage. The results of the study showed that compared to the control group (Figure 4A), DSS-exposed zebrafish exhibited tissue inflammation and major pathological changes, including villi hypertrophy, increased presence of vacuoles, reduction in goblet cells, deformation of columnar cells, and thickened muscle layer (Figure 4B). Similarly to DSS-induced lesions in mice, DSS-induced enteritis also led to intestinal changes in zebrafish. Furthermore, the Wgx-50 treatment group exhibited intact and well-arranged intestinal epithelial cell structure, along with normal glandular structure (Figure 4C), demonstrating a greater protective effect against DSS-induced alterations and suggesting the potential of Wgx-50 in preserving intestinal morphology. Measurements of villus length and crypt depth revealed that DSS treatment did not cause a significant change in villus length (Figure 4D). However, an increase in crypt depth, indicative of intestinal swelling, was observed and was subsequently alleviated by Wgx-50 treatment (Figure 4E). The reduction in the villus length to crypt depth ratio, a marker of enteritis, suggested the onset of DSS-induced enteritis, which was mitigated by Wgx-50 (Figure 4F). Further research is required to validate this finding and to explore the mechanisms by which Wgx-50 exerts its effects on intestinal health.

### 2.5. The Akt Pathway Mediates the Regulation of Inflammation by Wgx-50

Given the role of Akt in inflammation regulation, we hypothesize that Wgx-50 may modulate the expression of Akt in the zebrafish intestinal injury model to regulate the level of inflammation. Western blot analysis of Akt protein activity revealed that, compared to the control group, DSS treatment did not lead to significant changes in Akt protein levels, while the Wgx-50 treatment group showed a significant decrease in Akt protein levels (Figure 5B,C). qPCR analysis of Akt mRNA levels revealed a significant upregulation of *akt1*, *akt2*, and *akt3* in the DSS treatment group, while the expression levels of *akt1*, *akt2*, and *akt3* were downregulated after Wgx-50 treatment (Figure 5D–F). These results suggest that Wgx-50 may regulate the inflammatory process in the zebrafish intestinal injury model by downregulating Akt protein expression and RNA levels. Further research can explore in depth the regulatory mechanisms of Wgx-50 on the Akt signaling pathway and its role in inflammation regulation, thus enhancing our understanding of the potential mechanisms and therapeutic efficacy of Wgx-50 in treating intestinal inflammation and other related diseases.

## 3. Discussion

The inflammatory response plays a crucial role in the development of various diseases, including inflammatory bowel disease (IBD) [24]. As a chronic disease that debilitates individuals, IBD currently lacks a complete cure, and the available treatment methods have limitations including long-term medication and a considerable proportion of non-responders [25]. Therefore, exploring natural compounds with anti-inflammatory properties is of great significance for finding novel treatment strategies. Our study focuses on Wgx-50, a bioactive compound extracted from Sichuan pepper, which has gained considerable attention due to its anti-inflammatory potential [20]. Previous studies have indicated that Wgx-50 can exhibit anti-inflammatory effects. However, the role and underlying molecular mechanisms of Wgx-50 in DSS-induced colitis remain unclear [26].

Therefore, we used zebrafish as experimental animals to investigate the role of Wgx-50 in DSS-induced colitis by establishing an IBD model. In this regard, most IBD models utilize DSS, which leads to epithelial damage and inflammation [27]. The zebrafish colitis model has substantial advantages in simulating repetitive intestinal injury and repair [28]. The expression levels of cytokines to some extent reflect the level of inflammation, which can be used as indicators to evaluate the anti-inflammatory activity of natural compounds [23]. Pro-inflammatory cytokines are considered key participants in the initiation and development of IBD, with the most important ones being *IL-1β*, *IL-6*, *CXCL8a*, and *TNF-α* [23]. These inflammatory cytokines are associated with the two main clinical types of IBD (Crohn’s disease and ulcerative colitis) and are not limited to individual subsets of helper T cells [29,30].

First, we validated whether the inflammatory characteristics of the disease model we established were consistent with the DSS-induced zebrafish model of inflammatory bowel disease (IBD) that we referenced: We selected two inflammatory markers, neutrophil count and pro-inflammatory cytokine expression, to validate whether they showed an increase or upward trend. In this study, the transcription levels of *IL-1β*, *IL-6*, *CXCL8a*, and *TNF-α* were significantly increased in the DSS group, indicating increased expression of pro-inflammatory cytokines and recruitment of neutrophils by DSS (Figure 1). Additionally, we found that Wgx-50 could alleviate the expression of pro-inflammatory cytokines in the DSS-induced zebrafish model of colitis (Figure 2).

This result suggests that Wgx-50 may regulate the inflammatory response, thereby helping to improve colitis. Furthermore, we found that Wgx-50 inhibited the recruitment of neutrophils to the site of intestinal inflammation. Neutrophils are key immune cells involved in the initial response to tissue damage or infection, and their excessive activation and accumulation can lead to the worsening of inflammatory diseases [31]. The inhibition of neutrophil recruitment by Wgx-50 suggests its potential as an anti-inflammatory agent.

It has been reported that DSS-induced damage is due to the formation of luminal fatty acid complexes during DSS treatment, leading to a decrease in barrier function [11,32]. Chronic DSS models in mice have been reported, but they require several months to develop [33]. Abnormal intestinal morphology includes reduced epithelial proliferation, apoptosis of epithelial cells and goblet cells, severe inflammation, and disruption of the normal balance between bacteria and the intestine [34]. Our data show that DSS causes intestinal changes in zebrafish, such as increased luminal vacuoles, enlargement of small intestinal villi, and a reduction in the number of goblet cells. Treatment with Wgx-50 demonstrated a protective effect against DSS-induced colitis (Figure 4).

Akt is a serine/threonine kinase that plays a crucial role in regulating tumor cell growth and proliferation, promoting cell invasion and metastasis, stimulating angiogenesis, and contributing to chemotherapy and radiotherapy resistance [35]. It is a central node in many signaling pathways and also plays an important role in regulating inflammatory responses. We speculate that Wgx-50 may regulate inflammatory responses by inhibiting the Akt protein. Our research results elucidate that the Akt signaling pathway is involved in the modulation of the anti-inflammatory effects of Wgx-50 (Figure 5).

In conclusion, zebrafish share a large number of homologous genes with humans, and their genes related to inflammatory diseases are highly similar to those in humans. Additionally, the activity level of zebrafish provides a solid foundation for this study. Our study provides evidence for the potential application of Wgx-50 as a natural compound with immunomodulatory properties in colonic inflammation. By reducing the expression of pro-inflammatory cytokines and inhibiting the recruitment of neutrophils, Wgx-50 demonstrated its anti-inflammatory properties in the zebrafish DSS-induced colitis model. Furthermore, we found that the Akt signaling pathway is involved in the modulation of the anti-inflammatory effects of Wgx-50. Our study provides potential research directions for other inflammatory conditions related to Wgx-50, but further investigation is needed to elucidate the specific molecular mechanisms underlying the anti-inflammatory effects of Wgx-50 and its potential as a therapeutic target for IBD and other inflammatory diseases.

## 4. Materials and Methods

### 4.1. Zebrafish Strains and Maintenance

Wild-type (WT) and transgenic strain Tg (*lyz*:EGFP) zebrafish were used in the experiment, with the Tg (*lyz*:EGFP) strain expressing the fluorescent protein EGFP to label neutrophils [36]. According to previous studies, zebrafish embryos were obtained by mating with a sex ratio of 1:2 [36]. The obtained embryos were cultured under constant conditions of 28.5 °C, pH 6.5–7.5, with a light/dark cycle of 14 h/10 h [37]. Zebrafish breeding was carried out strictly according to the guidelines and regulations set by the Animal Resource Center and the university regulations of Anhui Agricultural University. Additionally, all experiments complied with the guidelines of the National Institutes for Food and Drug Control of China regarding the use of experimental animals.

### 4.2. Drug Treatments

Zebrafish embryos (3 dpf) were exposed to freshly prepared 0.3%, 0.4%, 0.5%, 0.6%, and 0.7% sodium dextran sulfate (DSS) (MACKLIN, China) for three days to induce intestinal inflammation. The effects on neutrophil migration and cytokine expression were observed. Wgx-50 (10 μL/L) was used to treat zebrafish with intestinal injury, and the effects on neutrophil migration and cytokine expression were observed at 6 h, 12 h, and 18 h.

### 4.3. Behavioral Monitoring

For behavioral analysis, the experiment was divided into control group, DSS-treated group, and Wgx-treated group, with a pre-treatment of 72 h for all groups. Each group consisted of 16 fish and was placed in a 48-well plate. The procedure of the PMR experiment was as follows: a 30 min adaptation to darkness was conducted, followed by alternating 5 min periods of light exposure and darkness, repeated three times. Finally, a 2 min dark interval was included to observe the activity of the larvae [38].

### 4.4. Histopathological Sections

The zebrafish intestines were fixed in 4% paraformaldehyde at 4 °C for 24 h. Dehydration of the zebrafish intestines was performed using a series of ethanol dilutions (gradually increasing ethanol concentrations, such as 30%, 50%, 70%, 80%, 90%, and a final concentration of 100% ethanol). Subsequently, the tissues were infiltrated with paraffin as a substitute for ethanol to allow gradual tissue penetration. Zebrafish intestinal sections that underwent fixation, dehydration, and paraffin embedding were obtained and stained with hematoxylin and eosin. Finally, the sections were mounted in a transparent medium for microscopic observation, and differences between different experimental groups were compared.

### 4.5. Vivo Imaging

Treatment of 3 dpf wild-type (WT) and transgenic strain Tg (*lyz*:EGFP) zebrafish with varying concentrations of DSS and Wgx (10 μL/L), followed by observation of the effects of DSS and Wgx on intestinal inflammation within 72 h post-treatment.

Live imaging of the juveniles was performed using a fluorescence microscope (Nexcope, China), and the aggregation of neutrophils in the intestinal region was observed by capturing images. The number of neutrophils in the intestinal region was analyzed using the ImageJ (Version 1.52a) software platform.

### 4.6. Quantitative Polymerase Chain Reaction

After treating zebrafish with DSS and Wgx for 72 h, 30 juvenile fish were collected per group and placed in 400 μL TRIZOL (Takara, 9108). Subsequently, the tissues were subjected to sonication, followed by chloroform and isopropanol extraction of total RNA as previously described [38,39].

The extracted total RNA was reverse transcribed into cDNA and diluted 10-fold with sterile distilled water for qPCR experiments. Quantitative real-time fluorescence qPCR (qPCR) was performed using SYBR Premix^®^ Ex Taq™ (Takara, RR430S) on a LightCycler^®^ 96 instrument (Roche). Three independent samples were tested in each experiment. The experimental results were presented as relative expression calculated using the 2^−ΔΔ^ct method [36,39].

### 4.7. Western Blotting

Protein samples from 50 zebrafish juveniles treated with DSS (0.5%) and Wgx (10 μL/L) were collected separately. The protein samples were separated by 12% SDS-PAGE gel electrophoresis and then transferred onto nitrocellulose membranes (Servicebio, China)**.** Incubation with primary antibodies, including Actin (HUABIO, MA5-32479), p-AKT (CST, 4060T), and AKT (CST, 4691T), was performed at room temperature for 2 h. These antibodies can recognize proteins from zebrafish. After washing, incubation with HRP-conjugated secondary antibody (HUABIO, HA1001) was performed at room temperature for 2 h, followed by chemiluminescence imaging. Three samples were used for each experiment. The grayscale values of the proteins were analyzed using the ImageJ software platform. The experimental results were expressed relative to Actin.

## 5. Conclusions

The GraphPad Prism 9.0 software platform was used to analyze the experimental data. Data were analyzed by one-way ANOVA and t-test, and expressed as mean ± standard deviation. ** p* < 0.05, ** *p* < 0.01, ***** p* < 0.001, ****** p* < 0.0001.

## Figures and Tables

**Figure 1 ijms-25-09510-f001:**
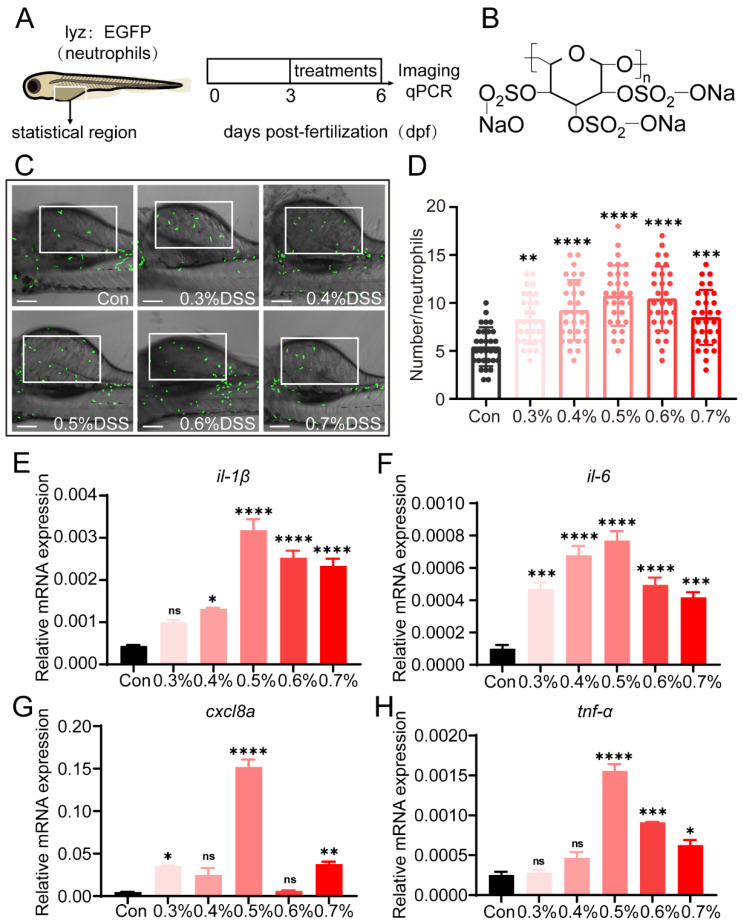
DSS promotes the recruitment of neutrophils and the expression of pro-inflammatory cytokines. (**A**) Tg(lyz:EGFP) zebrafish larvae (3 dpf) were exposed to different concentrations of DSS for three days. (**B**) Images display the chemical formula of DSS. (**C**) Images demonstrate the recruitment of neutrophils in the zebrafish intestinal injury model under fluorescence microscopy. White rectangles represent counting areas. Scale: 100 μm. (**D**) Shows the statistical quantification of neutrophil recruitment in the zebrafish intestinal injury model under control group and different concentrations of DSS treatment. The recruitment of neutrophils in the zebrafish intestine increases to varying degrees with different concentrations of DSS treatment. (**E**–**H**) Display the gene expression levels of pro-inflammatory cytokines il-1, il-6, cxcl8a, and tnf-α in zebrafish under control group and different DSS concentrations. The levels of pro-inflammatory cytokines in zebrafish increase to varying degrees with different concentrations of DSS treatment. All statistical analyses are compared to the control group (Con group). (ns, not significant; * *p* < 0.05, ** *p* < 0.01, *** *p* < 0.001, **** *p* < 0.0001 one-way ANOVA.)

**Figure 2 ijms-25-09510-f002:**
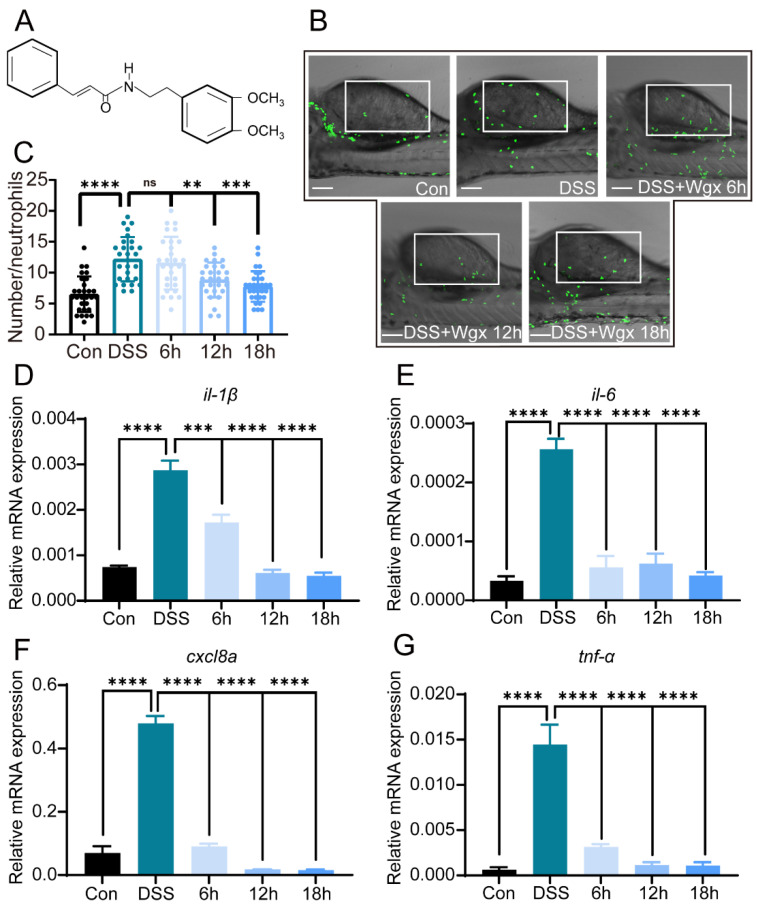
Wgx-50 inhibits the recruitment of neutrophils and the expression of pro-inflammatory cytokines induced by DSS. (**A**) Images display the chemical formula of Wgx-50. (**B**) Images demonstrate the recruitment of neutrophils in the zebrafish intestinal injury model under fluorescence microscopy. White rectangles represent counting areas. Scale: 100 μm. (**C**) The statistical quantification of neutrophil recruitment in zebrafish intestinal injury model under control group, DSS group, and DSS + Wgx-50 group. The recruitment of neutrophils in the zebrafish intestine increases after DSS treatment, and the number of neutrophil recruitment gradually decreases with the addition of Wgx-50 over time. (**D**–**G**) The gene expression levels of pro-inflammatory cytokines il-1, il-6, cxcl8a, and tnf-α in zebrafish under control group, DSS group, and DSS + Wgx-50 group. The levels of pro-inflammatory cytokines increase after DSS treatment, and the levels gradually decrease with the addition of Wgx-50 over time. The primer sequences are listed in Table 1. (ns, not significant;, ** *p* < 0.01, *** *p* < 0.001, **** *p* < 0.0001, one-way ANOVA.)

**Figure 3 ijms-25-09510-f003:**
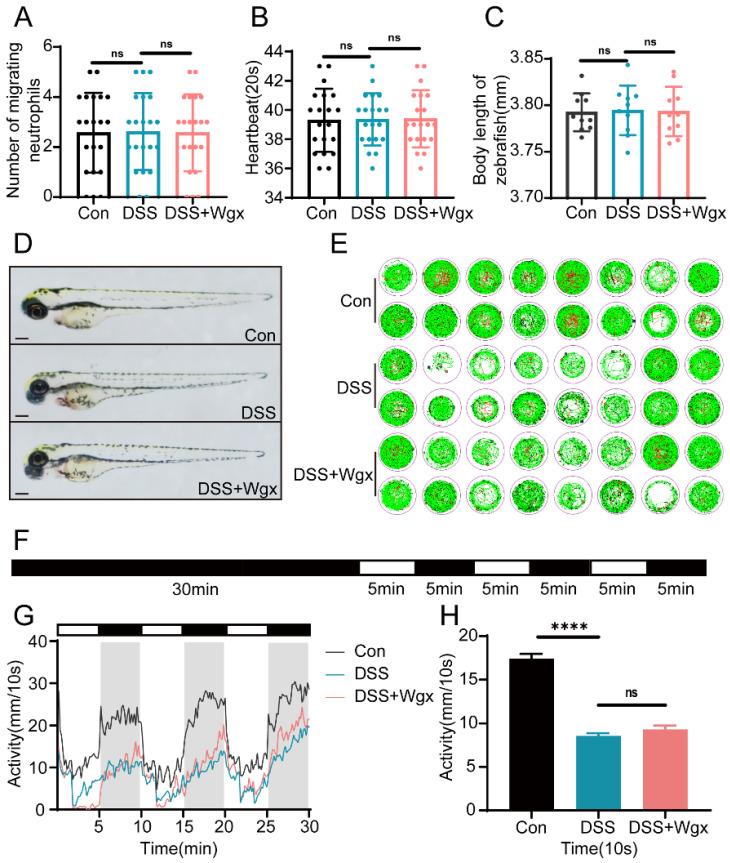
DSS and Wgx-50 have no effect on the growth and development of zebrafish. (**A**) The number of spontaneous movements per minute in zebrafish larvae at 30 hpe (*n* = 20). There was no significant effect observed after drug treatment. (**B**) The number of heartbeats every 20 s in zebrafish larvae at 48 hpe (n = 20). There was no significant effect observed after drug treatment. (**C**) The body length of zebrafish larvae at 96 hpe (n = 10/group). There was no significant effect observed after drug treatment. (**D**) The morphology of zebrafish at 96 hpf was captured using a stereomicroscope. Scale bar: 100 μm. (**E**) Zebrafish larval PMR behavior monitored by Viewpoint system (n = 16). (**F**) The experimental workflow of PMR is shown in the diagram. (**G**,**H**) Data analysis of zebrafish larval PMR behavior monitored by Viewpoint system (n = 16). There was a significant decrease in activity after drug treatment. (ns, not significant; **** *p* < 0.0001, one-way ANOVA.)

**Figure 4 ijms-25-09510-f004:**
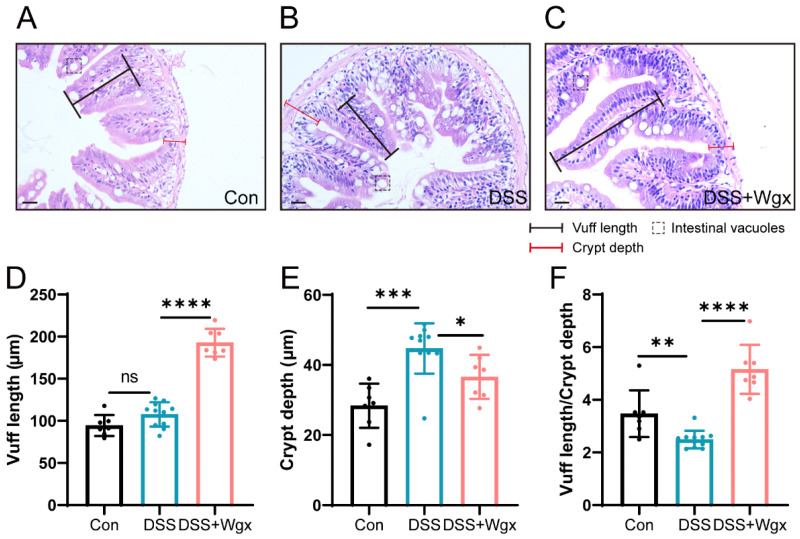
Protective effect of Wgx-50 on intestinal morphology in zebrafish after DSS treatment. (**A**) H&E staining of control group zebrafish intestinal sections shows normal tissue structure. (**B**) Zebrafish intestinal sections exposed to DSS show evident tissue inflammation and pathological changes. (**C**) Zebrafish intestinal sections treated with Wgx show a protective effect. Scale bar: 20 μm. (**D**–**F**) Zebrafish intestinal villus length, crypt depth, and the ratio of villus length to crypt depth in the Con, DSS, and DSS + Wgx groups. (ns, not significant; * *p* < 0.05, ** *p* < 0.01, *** *p* < 0.001, **** *p* < 0.0001, one-way ANOVA.)

**Figure 5 ijms-25-09510-f005:**
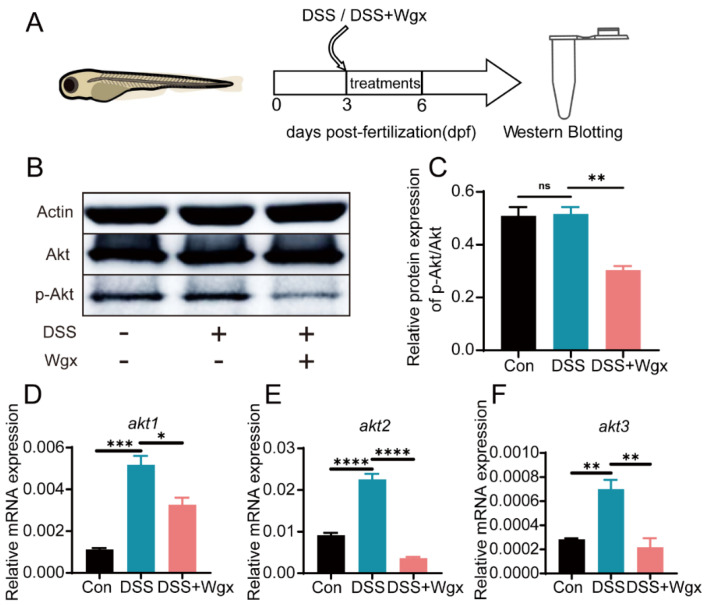
Wgx-50 affects inflammation by regulating Akt expression. (**A**) Zebrafish larvae (3 dpf) were exposed to DSS or DSS+Wgx on the third day and treated for three days, followed by Western blot analysis. (**B**,**C**) Protein expression levels of p-Akt/Akt in control group, DSS group, and DSS + Wgx group are shown. There was no significant change in p-Akt/Akt protein expression levels after DSS treatment, while the protein levels significantly decreased after Wgx treatment. (**D**–**F**) Gene expression levels of akt1, akt2, and akt3 in control group, DSS group, and DSS + Wgx group are shown. There was a significant upregulation of akt1, akt2, and akt3 gene expression levels after DSS treatment, while the gene levels significantly decreased after Wgx treatment. (ns, not significant; * *p* < 0.05, ** *p* < 0.01, *** *p* < 0.001, **** *p* < 0.0001, one-way ANOVA.)

**Table 1 ijms-25-09510-t001:** Primers designed for expression analysis in this study.

Gene	Gene ID	Note	Forward Primer (5′-3′)	Reverse Primer (5′-3′)
tnf-α	NM_212859.2	qRT-PCR	GCGCTTTTCTGAATCCTACG	TGCCCAGTCTGTCTCCTTCT
il-1β	NM_212844.2	qRT-PCR	GTACTCAAGGAGATCAGCGG	CTCGGTGTCTTTCCTGTCCA
il-6	NM_001261449.1	qRT-PCR	GCTATTCCTGTCTGCTACACTGG	TGAGGAGAGGAGTGCTGATCC
cxcl8a	XM_009306855.3	qRT-PCR	CCACACACACTCCACACACA	CCACTGAATTGTCCTTTCATCA
β-actin	NM_131031.2	qRT-PCR	ACGAACGACCAACCTAAACTCT	TTAGACAACTACCTCCCTTTGC
akt1	NC_007128.7	qRT-PCR	GTTTTCTGCGGGATTTCAGCG	GTCTTCACACGGGTCACCAGG
akt2	NC_007129.7	qRT-PCR	AAAGCTGCTGGGTAAAGGCA	TGCAACTTCGTCCTTAGCGA
akt3	NC_007124.7	qRT-PCR	AATAACCCCTCCTGAAAAATTTGAT	CGAGTAGGAGAACTGGGGGA

## Data Availability

Data contained within the article.

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
