# Peer review of "Lemairamin (Wgx-50) Attenuates DSS-Induced Intestinal Inflammation in Zebrafish"

_ijms, 2024, doi:10.3390/ijms25179510_

Round 1

Reviewer 1 Report

Comments and Suggestions for Authors

This paper by Zhang and collaborators reports a set of interesting results regarding the protective effects of Wgx-50 (piperine) in a zebrafish model of DSS-induced intestinal inflammation. By using a combination of histochemical, beahvioural, biochemical and molecular techniques, the Authors show that Wgx-50 reduces the expression of pro-inflammatory cytokines induced by DSS and inhibit the recruitment of neutrophils to the site of intestinal injury. Also, they show that Wgx-50 exerts its anti-inflammatory effect by regulating the activation of the Akt pathway. 

Major points

None

Minor points

- The Authors could extend the 'Introduction' section in order to make it clear how the DSS-induced intestinal inflammation model they propose has been used to generate relevant results in the study of molecules other than Wgx-50. 

- The Authors should extend the 'Discussion' section in order to explain the advantages of using their DSS-induced zebrafish model in the analysis of Wgx-50 effects and to make the reader understand that their model has similarly been used to study the effect of other molecules.  

- The Authors should revise the format of the 'References' section thoroughly since it does not respect the indications of the journal.

Author Response

Thank you very much for your review and valuable comments.

Research has shown that grape exosome-like nanoparticles can induce intestinal stem cells and protect mice from DSS-induced colitis, which also demonstrates the feasibility of using DSS to induce colitis. We have cited this conclusion in our introduction.

Additionally, we have added a discussion on the advantages of using zebrafish for studying DSS-induced colitis models. Zebrafish have high genetic homology with humans in inflammation-related diseases and offer convenient advantages for behavioral studies.

We have also noted your comments regarding the format of the references and have made the necessary corrections.

Reviewer 2 Report

Comments and Suggestions for Authors The main question addressed in the study is to assess the effect of piperine Wgx-50 on dextran sulfate sodium-induced colitis inflammation. This study is original in terms of investigating the Wgx-50 in zebrafish.
The study demonstrates the inflammatory bowel disease in zebrafish.

1. The title may be revised to explain the meaning of Wgx-50.

2. Figure 4 needs some explanation to indicate the protective effect in the figure.

3. It is not quite clear what Wgx-50 means and the difference between piperine and Wgx-50. The authors may explain more detailed explanation for the difference between piperine and Wgx-50 in Introduction.

4. Figure 1 needs some improvements in the significance visualization with asterisks. it is not sure whether what bars were compared to calculate p value in Figures D-H.

5. The protective effect of Wgx-50 needs to be highlighted.

6. Figure 5 needs improvements to highlight that the experiments were performed in zebrafish model.

7. Some explanations whether the antibodies for p-AKT and AKT recognize the zebrafish proteins are needed in Western Blotting in Materials and Methods section.

8. In Discussion, some description that the experiments in the study were performed in zebrafish and the effects in human are unknown is needed.

Author Response

Comments 1: The title may be revised to explain the meaning of Wgx-50.

Response 1 : Thank you for your guidance. To clarify that Wgx-50 is an extract of Lemairamin, we have updated the title to: 'Lemairamin (Wgx-50) attenuates DSS-induced intestinal inflammation in zebrafish.

Comments 2: Figure 4 needs some explanation to indicate the protective effect in the figure.

Response 2 : Thank you for your guidance. We have revised Figure 4 to show the measured villus length and crypt depth data, which helps to better explain the protective effect of Wgx-50 against DSS-induced intestinal inflammation.

Comments 3. It is not quite clear what Wgx-50 means and the difference between piperine and Wgx-50. The authors may explain more detailed explanation for the difference between piperine and Wgx-50 in Introduction.

Response 3 : Thank you very much for the valuable feedback on our article. Piperine is an extract derived from pepper and chili, while Wgx-50 is an extract from Sichuan pepper and originates from Lemairamin. We have revised the explanation of Wgx-50 in the text and the title to clarify that Wgx-50 is an extract of Lemairamin.

Comments 4. Figure 1 needs some improvements in the significance visualization with asterisks. it is not sure whether what bars were compared to calculate p value in Figures D-H.

Response 4: Thank you for your feedback on our article. We want to express that each group is compared individually to the Con group. We have now revised Figures 1D-H and clarified in the figure legend that the results shown at the top of the bars are compared to the Con group.

Comments 5: The protective effect of Wgx-50 needs to be highlighted.

Response 5: Thank you very much for your valuable feedback. To facilitate a better understanding of the protective effect of Wgx-50, we have presented data on the ratio of villus length to crypt depth, and have marked the vacuolar areas of the villi in the figure. We have also provided an explanation in the text, noting that a decreased ratio of villus length to crypt depth is indicative of DSS-induced intestinal inflammation, and that Wgx-50 can restore this value. This should help readers better understand the protective effect of Wgx-50 on inflammation and the changes in intestinal status.

Comments 6. Figure 5 needs improvements to highlight that the experiments were performed in zebrafish model.

Response 6: Thank you very much for your guidance. We have added a schematic diagram to Figure 5 to illustrate the experimental protocol, indicating that our experiments were conducted in zebrafish.

Comments 7. Some explanations whether the antibodies for p-AKT and AKT recognize the zebrafish proteins are needed in Western Blotting in Materials and Methods section.
Response7: Thank you very much for your guidance. We have provided the catalog numbers for the reagents used in the Materials and Methods section and have also added a note in the text specifying that p-AKT and AKT antibodies can be used to detect zebrafish proteins.

Comments 8: In Discussion, some description that the experiments in the study were performed in zebrafish and the effects in human are unknown is needed.

Response8: Thank you very much for your review and guidance. Our main work was conducted using a zebrafish model, which shares high genetic homology with humans in inflammation-related diseases. We have added this explanation in the discussion section.